# Adaptive Data Fusion Method of Multisensors Based on LSTM-GWFA Hybrid Model for Tracking Dynamic Targets

**DOI:** 10.3390/s22155800

**Published:** 2022-08-03

**Authors:** Hao Yin, Dongguang Li, Yue Wang, Xiaotong Hong

**Affiliations:** 1School of Mechatronical Engineering, Beijing Institute of Technology, Beijing 100081, China; yoon@bit.edu.cn (H.Y.); lidongguang@bit.edu.cn (D.L.); hong@bit.edu.cn (X.H.); 2School of Mechatronical Engineering, North University of China, Taiyuan 038507, China

**Keywords:** multisensor, data fusion, LSTM, dynamic target tracking

## Abstract

In preparation for the battlefields of the future, using unmanned aerial vehicles (UAV) loaded with multisensors to track dynamic targets has become the research focus in recent years. According to the air combat tracking scenarios and traditional multisensor weighted fusion algorithms, this paper contains designs of a new data fusion method using a global Kalman filter and LSTM prediction measurement variance, which uses an adaptive truncation mechanism to determine the optimal weights. The method considers the temporal correlation of the measured data and introduces a detection mechanism for maneuvering of targets. Numerical simulation results show the accuracy of the algorithm can be improved about 66% by training 871 flight data. Based on a mature refitted civil wing UAV platform, the field experiments verified the data fusion method for tracking dynamic target is effective, stable, and has generalization ability.

## 1. Introduction

In the field of future air combat, high-precision tracking and monitoring of enemy aircraft by UAVs will be an indispensable link in the OODA combat ring, which provides important support for combat decision-making [1]. It requires the use of computer technology to analyze and synthesize the observed data of sensors to obtain a more accurate trajectory of targets. With the rapid development of modern aerospace technology, the speed, angle, acceleration, and other parameters of maneuvering targets in space are changing constantly, which makes the sequential position of dynamic targets have strong correlation. The research of tracking dynamic targets with maneuvering characteristics has become one hot spot of electronic warfare and there is an urgent need to study superior tracking filtering methods. Although the increased hardware performance of various sensors has slightly improved the accuracy of systems, radar and photoelectric sensors have still been used to track enemy aircraft in recent decades. Moreover, the significant impact of errors of the measurement system on the distance and angle of sensors should not be underestimated. In order to satisfy the accuracy of tracking enemy aircraft in the process of moving, an adaptive online multisensor data fusion method is needed to improve the traditional Kalman filter (KF) and other algorithms. Multisensor data fusion (MSDF) is making full use of multisource data and combining their redundancy or complementarity according to rules based on the hardware of multisensor systems, so as to obtain a more essential and objective understanding of the same thing. The tracking trajectory of dynamic targets is based on updating the system states in each time-step by fusion of multisensor results, which can be more robust than any single data. A novel architecture of MSDF can give the system more accurate estimation of error in each KF-based model to obtain better fusion results. In this paper, a new data fusion method is proposed which integrates a Kalman filter, least squares (LS), a neural network (NN), and the maneuver detection mechanism to improve the performance of filtering. In contrast to traditional data fusion, which only considers a single filtering process, this method focuses on the architecture of global fusion, which makes full use of the relationship between the combined and the individual multisensors. It combines the impact of historical data on the current data, so as to realize online and real-time adaptive, fast, and accurate data fusion.

In the task of tracking enemy aircraft for air combat, this research used a data fusion scene composed of UAVs with sensors and enemy aircraft, as illustrated in Figure 1. There was a complete data link between UAVs which could communicate with each other at any time. Each UAV carried only one type of detection sensor (photoelectric or radar sensor). Enemy aircraft attempted to escape tracking through various maneuvers, causing errors in the continuous detection of their position by UAVs. When the sensor detection of more than two UAVs obtained the position of the enemy aircraft, data fusion could be carried out and the final fusion results would be shared to realize the cooperative tracking of multiple UAVs. Then the UAVs with lost tracking field of view could be redirected to the correct tracking path. The research content of this paper is the process of autonomous data fusion after the UAVs obtained the multisource state of enemy aircraft using multisensors.

To solve the problem of data fusion of multisensors for tracking of enemy aircraft, this paper puts forward several assumptions.

A.There may be weak communication delay between UAVs. The acquired position data error of enemy aircraft caused by communication delay of UAVs can be corrected by time and space alignment. (This paper focused on the data fusion process, rather than how to reduce the communication delay, so the pre-processing of data is not what the fusion algorithm itself needed to consider).B.Each UAV detects the target independently, so the position data errors of enemy aircraft obtained by each sensor are irrelevant. (In the practical scenario considered in this paper, the sensors carried by UAVs did not provide cooperation and support for other sensor detection. Therefore, the interaction of data errors was ignored).C.The relative motion of UAVs and enemy aircraft can be regarded as in the same plane during the continuous tracking process in discrete time. (For any maneuver form of escape aircraft, the UAV adopt the form of bank-to-turn (BTT) and keep it in the same relative plane, rather than the horizontal plane and the same height).D.Once the enemy aircraft takes maneuvering flight, its main states of speed and direction will change. (Dynamic changes of enemy aircraft are inevitable. Speed and direction directly lead to position changes, which is very important for occupancy guidance or defensive games in air combat. Sometimes acceleration may be considered).

The purpose of existing data fusion is to eliminate the uncertainty in measurement and obtain more accurate and reliable measurement results than the arithmetic mean of limited measurement information [2], since there are many methods of data fusion, such as weighted average, classical organon, Bayesian estimation, and so on.

This paper used global filtering and a long short-term memory (LSTM) network to improve measurement error prediction of a least squares Kalman filter as the basic fusion method. It used the statistical characteristics of the measurement model to determine the optimal estimation of the fusion data recursively. A general data fusion method for tracking dynamic target was constructed to realize real-time high-precision trajectory acquisition, verified by numerical simulation and field test.

The contributions of this paper include: (1) Based on the traditional multisensor weighted fusion algorithm, a new data fusion method using a global Kalman filter and the predicted measurement variance of LSTM was designed. An adaptive truncation mechanism was used to determine the optimal weights by considering the influence of multisensor data fusion on a single datum. (2) A maneuvering detection mechanism was added through a variable dimension Kalman filter, whose results were used as data fusion parameters. The effectiveness and stability of the fusion method for dynamic tracking were verified by numerical simulation and field experiments and it has certain generalization ability.

The rest of the paper is organized as follows. Section 2 presents a complete review of prior works in the literature relevant to the research. Section 3 describes the preparatory work of data fusion, and proposes a new architecture of weighted data fusion based on a Kalman filter, which makes the fusion result error smaller and closer to the real value. Section 4 analyzes and discusses the results of numerical simulation and field tests. Finally, Section 5 draws conclusions and outlines future research directions.

## 2. Related Works

As an important data processing method, data fusion technology has a wide range of applications, such as target detection [3,4], battlefield evaluation [5], medical diagnosis [6], remote sensing mapping [7,8,9], fault diagnosis [10,11,12], intelligent manufacturing [13,14,15], etc. Traditional data fusion methods can be divided into groups based on probability, Dempster Shafer theory, knowledge, etc. [16].

The data fusion methods based on probability include Bayesian inference [17], the Kalman filter model [18,19], and the Markov model [20]. They represent the dependence between random variables by introducing probability distribution and probability density function. Sun et al. [21] proposed a weighted fusion robust incremental Kalman filtering algorithm and introduced the incremental formula to eliminate the unknown measurement error of the system. Xue et al. [22] proposed an adaptive multi-model extended Kalman filter (AME-EKF) estimation method based on speed and flux which can adjust the transition probability and system noise matrix continuously and adaptively to reduce the speed estimation error by using residual sequence. Huang et al. [23] proposed a weighted fusion method of real-time separation and correction of outliers which solved the problem of the decline of ballistic data processing accuracy caused by the pollution of measurement data by introducing the robust estimation theory. Aiming at the problem that the classical interactive multi-model (IMM) cannot provide the a priori information of the target motion model, Xue et al. [24] proposed a multisensor hierarchical weighted fusion algorithm for maneuvering target tracking which obtained its model through hierarchical weighting, and estimated the state using an extended Kalman filter (EKF).

The fusion method based on Dempster Shafer (D-S) theory expresses the uncertainty of data by introducing confidence and rationality in dynamic situations. In order to avoid counter-intuitive results when combining highly conflicting pieces of evidence, Xiao et al. [25] proposed a weighted combination method for conflicting pieces of evidence in multisensor data fusion by considering both the interplay between the pieces of evidence and the impacts of the pieces of evidence themselves [sensors]. Zhao et al. [26] proposed a multiplace parallel fusion method based on D-S evidence theory for sensor network target recognition information fusion in which each sensor only needs to exchange information with neighbor sensors. Aiming at the limitations of simple and incomplete information sources of single sensor fault diagnosis, Zheng et al. [27] proposed a multisensor information fusion fault diagnosis method based on empirical mode decomposition sample entropy and improved D-S evidence theory to realize the effective fusion of multiple sensor information and the information complementarity between sensors is formed to achieve the purpose of optimal decision-making.

The fusion method based on knowledge mainly uses support vector machine and clustering methods to find the knowledge contained in the data and measure the correlation and similarity between knowledge. Guan [28] proposed a data fusion method of high-resolution and multispectral remote sensing satellite images based on multisource sensor ground monitoring data by introducing the theory and method of support vector machine (SVM) and applied it to water quality monitoring and evaluation. Zhu et al. [29] proposed a hybrid multisensor data fusion algorithm based on fuzzy clustering by designing a strategy to determine the initial clustering center, which can deal with the limitations of traditional methods in data with noise and different density.

Since the proposal of deep learning, its application in data fusion has attracted the attention of a large number of researchers. In order to improve the accuracy of wireless sensor network data fusion, Cao et al. [30] proposed a wireless sensor network data fusion algorithm (IGWOBPDA) based on an improved gray wolf algorithm and an optimized BP neural network by improving control parameters and updating the position of dynamic weights. To improve the performance of data fusion in wireless sensor networks, Pan [31] proposed a data fusion algorithm that combines a stacked autoencoder (SAE) and a clustering protocol based on the suboptimal network powered deep learning model. Aiming at the problem that soft sensing modeling methods of most complex industrial processes cannot mine the process data resulting in low prediction accuracy and generalization performance, Wu [32] proposed a soft sensing method based on an extended convolutional neural network (DCNN) combined with data fusion and correlation analysis. However, most data fusion research is the application of traditional filtering architecture to seek fewer errors and accurate target state estimation in the field of tracking. Yu et al. [33] reviewed the LSTM cell and its variants to explore the learning capacity of the LSTM cell and proposed future research directions for LSTM networks. Sherstinsky [34] explained the essential RNN and LSTM fundamentals in a single document, which formally derived the canonical RNN formulation from differential equations by drawing from concepts in signal processing. Ye et al. [35] proposed an attention-augmentation bidirectional multiresidual recurrent neural network (ABMRNN), which integrates both past and future information at every time step with an omniscient attention model. Duan et al. [36] proposed a tracking method for unmarked AGVs by using the frame difference method and particle filter tracking, then using a support vector machine (SVM) model to predict the area and correct the prediction results of the LSTM network.

## 3. Methodology

### 3.1. Least Square Fitting Estimation

The weighted data fusion algorithm uses the least squares criterion to minimize the weighted error sum of squares, when n sensors track a dynamic target at the same time. The weighted least squares estimation criterion is to make the sum of the weighted error squares take the minimum value and obtain the weighted least squares estimate by finding the minimum value [37]. In the process of multisensor cooperative detection, the observation of a certain state parameter of the detected target by n sensors is as shown in Equation (1).
(1)y=Hx+ewhere x is a one-dimensional measured vector, which represents the true value of a certain parameter of the detected target; y is an n-dimensional measurement vector, which is a vector composed of multiple sensor observations of a certain parameter of the target, set $y={\left[y_{1}\;y_{2}\;\cdots \;y_{n}\right]}^{T}$; H is a known n-dimensional constant vector under ordinary conditions; e is an n-dimensional vector composed of the measured noise of each sensor including the internal noise of the sensor system and environmental interference noise, set $e={\left[e_{1}\;e_{2}\;\cdots \;e_{n}\right]}^{T}$.

The criterion of least square estimation is to minimize the sum of squares of weighted errors $T\left(\hat{x}\right)$.
(2)$$T\left(\hat{x}\right)={\left(y-H\hat{x}\right)}^{T}W\left(y-H\hat{x}\right)$$
where W is a positive definite diagonal weighted matrix, which represents the fusion weight of each sensor, set $W=\mathrm{diag}\left[w_{1}\;w_{2}\;\cdots \;w_{n}\right]$. $\hat{x}$ is the estimated value of the true value x. In order to ensure the minimum value of Equation (2), calculate the partial derivative to obtain Equation (3).
(3)$$\frac{\partial T\left(\hat{x}\right)}{\partial \hat{x}}=-H^{T}\left(W+W^{T}\right)\left(y-H\hat{x}\right)$$

Make the reciprocal of Equation (3) be zero to obtain the estimated value $x_{fusion}$. The least square estimate of the method is as shown in Equation (4).
(4)$$x_{fusion}={\left(H^{T}WH\right)}^{-1}\;H^{T}\;W\;y=\frac{\sum_{i=1}^{n}w_{i}y_{i}}{\sum_{i=1}^{n}w_{i}}$$

### 3.2. Fusion Weights and Measurement Variance

The noise of the sensor changes randomly and is irrelevant at any time. Therefore, the following assumptions can be made for the measurement noise of sensors: ① The measurement noise of each sensor is an ergodic signal with each state. ② The measurement noise outputs of each sensor are Gaussian white noise, which obey normal distribution and are independent of each other.
(5)$$E\left(e_{i}\right)=0\;\left(i=1,2,3\ldots n\right)$$
(6)$$E\left(e_{i}{}^{2}\right)=E\left[{\left(z_{i}-x\right)}^{2}\right]=\sigma_{i}{}^{2}\;\left(i=1,2,3\ldots n\right)$$
(7)$$E\left[\left(x-z_{i}\right)\left(x-z_{j}\right)\right]=0\;\left(i\neq j\right)$$
where $z_{i}$ or $z_{j}$ is a observed value of a sensor. According to the measurement noise characteristics of each sensor, mean error $\tilde{x}$ of the unbiased estimator $\hat{x}$ at the target of a state parameter x is calculated as shown in Equation (8).
(8)$$\tilde{x}=E\left[{\left(x-\hat{x}\right)}^{2}\right]=E\left[{\left(\frac{\sum_{i=1}^{n}w_{i}x-\sum_{i=1}^{n}w_{i}z_{i}}{\sum_{i=1}^{n}w_{i}}\right)}^{2}\right]$$

Equation (8) is calculated and expanded as shown in Equations (9) and (10).
(9)$${\tilde{x}}_{1}=E\left\{\sum_{i=1}^{n}\left[{\left(\frac{w_{i}}{\sum_{i=1}^{n}w_{i}}\right)}^{2}{\left(x-z_{i}\right)}^{2}\right]\right\}$$
(10)$${\tilde{x}}_{2}=E\left\{\sum_{i=1}^{n}\sum_{j=1,j\neq i}^{n}\left[\frac{w_{i}y_{i}}{{\left(\sum_{i=1}^{n}w_{i}\right)}^{2}}\right]\left(x-z_{i}\right)\left(x-z_{j}\right)\right\}$$

Equation (7) indicates that the value $\tilde{x}$ of Equation (10) is zero, so the mean error value of unbiased estimator $\hat{x}$ of target state parameter x is as shown in Equation (11).
(11)$$\tilde{x}=E\left[{\left(x-\hat{x}\right)}^{2}\right]=E\left[{\left(\frac{\sum_{i=1}^{n}w_{i}x-\sum_{i=1}^{n}w_{i}z_{i}}{\sum_{i=1}^{n}w_{i}}\right)}^{2}\right]=E\left\{\sum_{i=1}^{n}\left[{\left(\frac{w_{i}}{\sum_{i=1}^{n}w_{i}}\right)}^{2}{\left(x-z_{i}\right)}^{2}\right]\right\}$$

In order to find the minimum value of Equation (11), take the partial derivative of $w_{i}$ and make it equal to zero. Then, the relationship between fusion weight $w_{i}$ and measurement variance $\sigma_{i}{}^{2}$ is obtained as shown in Equation (12).
(12)$$w_{i}=\frac{1}{\sigma_{i}{}^{2}}$$
where $w_{i}$ is the weighted fusion coefficient of each sensor. $\sigma_{i}{}^{2}$ is the measurement variance of each sensor. $e_{i}$ is the noise measured by each sensor. $y_{j}$ is the observation value of each sensor. x is the real value of a state parameter of the target. $\hat{x}$ is the unbiased estimator of a state parameter x of the target. $\tilde{x}$ is the mean error between the estimated value $\hat{x}$ and the real value x. ${\tilde{x}}_{1}$ and ${\tilde{x}}_{2}$ are two parts of mean error $\tilde{x}$.

The derivation shows that the fusion weights of each sensor are only determined by measurement variance. The accuracy of sensor measurement variance calculation will affect the accuracy of data fusion results directly. So the accuracy of weights is the core to solve the problem of multisensor data fusion. The innovative algorithm of data fusion proposed in this paper is based on weighted data fusion, and the accuracy of measurement variance estimation will affect the effect of trajectory tracking ultimately.

### 3.3. Multisensor Fusion Architecture

In traditional data fusion, the original data is averaged directly as the input of the Kalman filter or extended Kalman filter to estimate the state of the target, which ignores the independence of each sensor and does not weaken the error in essence. In the least squares weighted data fusion algorithm, the final results of state estimation depend on the weight’s allocation according to the real-time measurement variance after the Kalman filter. There are more advanced methods of real-time measurement variance estimation, such as the forgetting factor, ordered weighted averaging (OWA) operator, and so on. However, they do not consider the correlation of sensor measurement variance in continuous time and the improvement of fusion architecture.

Based on the least square weighted Kalman filtering method, this paper proposes an adaptive multisensor weighted fusion algorithm with global Kalman filter and long short-term memory (LSTM-GWFA) network for tracking. The maneuver detection mechanism is introduced as the correction of the filtering results and LSTM is applied to the prediction of sensor measurement variance to improve the accuracy of weights allocation. The method is divided into four stages as shown in Figure 2: independent Kalman filter, global Kalman filter, measurement variance estimation, and data weighted fusion.

The stage of independent Kalman filtering works to filter the data of state obtained by the sensor on each independent mechanism, which can modify the prediction model and obtain the estimated data of the sensor itself.

The stage of global Kalman filtering works to obtain an initial fusion data from the independently filtered data of each sensor through a unified filter. The measurement variance of each sensor in the global filtering is calculated as the input of the next stage.

The stage of measurement variance estimation uses the measurement variance of sensor history as the input of a LSTM network to obtain the estimated measurement variance. Further, the estimated measurement variance is combined with the real-time measurement variance to obtain the final measurement variance of each sensor.

The stage of weighted fusion allocates the weights according to the final measurement variance of each sensor and to fuse the filtered data, then the result is sent back to the Kalman filter of each sensor to modify the prediction model.

#### 3.3.1. Independent Kalman Filter

Generally, there are two methods to obtain the state of aircraft under the research background of this paper. One is to measure it directly and obtain the value of observation. Another method is to predict it according to a certain rule based on the state of the previous moment. Due to the existence of observation noise, the inaccuracy of the prediction model and the estimation error of the position at the previous time, the two methods cannot obtain the available accurate estimation of state. The traditional weighted fusion method based on a Kalman filter combines the two results by weighting according to the values of noise covariance, state estimation covariance, and state transition covariance to obtain a relatively accurate state. Evolutionarily, this paper uses a variable dimension Kalman filter to switch constant velocity (CV) and constant acceleration (CA) models for data processing according to the motion characteristics of aircraft. In this stage, the maneuver is regarded as the internal change of the dynamic characteristics of the target rather than the noise modeling. A sliding window is used to count continuously the significant characteristics of the change of velocity and acceleration over a period of time, then switch the corresponding high-dimensional and low-dimensional models by determining whether the aircraft has maneuvering displacement according to the set threshold and confidence interval referred to in Appendix A.

The traditional Kalman filter uses the results of its own filtering as the input of the next prediction to correct its accuracy iteratively. However, whether this method will inevitably lead to the error correction depends on the measurement and filtering results at the current time in multisensor data fusion, and the filtering results of each sensor have no contribution to other prediction models. Therefore, this paper uses the final fusion results as the inputs of the next independent Kalman filter prediction model, which can modify the Kalman filter prediction model of each sensor accurately. The following Equations (13)–(18) show the implementation process of the independent Kalman filter.
(13)$$z_{i_{k}}=Hx_{i_{k}}+v_{i_{k}}$$
(14)$${\hat{x}}_{i_{k}}^{\prime }=Ax_{fusion}^{k-1}+Bu_{i_{k-1}}+o_{i_{k-1}}$$
(15)$$P_{i_{k}}^{\prime }=AP_{i_{k-1}}A^{T}+Q$$
(16)$$K_{i_{k}}=P_{i_{k}}^{\prime }H^{T}{\left[HP_{i_{k}}^{\prime }H^{T}+R\right]}^{-1}$$
(17)$${\hat{x}}_{i_{k}}={\hat{x}}_{i_{k}}^{\prime }+K_{i_{k}}\left(z_{i_{k}}-H{\hat{x}}_{i_{k}}^{\prime }\right)$$
(18)$$P_{i_{k}}=\left(I-K_{i_{k}}H\right)P_{i_{k}}^{\prime }$$
where $z_{i_{k}}$ is the observed value by sensor i at time k. ${\hat{x}}_{i_{k}}^{\prime }$ is the priori value of state estimate. ${\hat{x}}_{i_{k}}$ is the estimate value of posteriori state. A is the state transition matrix and B is the control matrix. H is the observation matrix. $u_{i_{k-1}}$ is the control variable. $v_{i_{k}}$ is the observation noise. $o_{i_{k-1}}$ is the noise of the prediction process. $P_{i_{k}}$ and $P_{i_{k-1}}$ are the posteriori estimated covariance. $P_{i_{k}}^{\prime }$ is the a priori estimated covariance. Q and R are measurement noise covariance. $K_{i_{k}}$ is the Kalman gain.

Combined with the fusion results $x_{fusion}^{k-1}$ of the previous moment and state transition matrix A, this paper used the estimation characteristics of the Kalman filter to filter the observed data $z_{i_{k}}$ of each sensor.

#### 3.3.2. The Global Kalman Filter

Through the in-depth study of the traditional algorithm of weighted data fusion, this paper finds that two defects lead to the low accuracy of the fusion results and lack of stability or anti-interference ability. Once the traditional algorithm is used to fuse the data of two sensors, the final fusion weight coefficient of each sensor is proved always to be 0.5 by mathematical theory. The performance of the algorithm is reduced greatly; also the algorithm is suitable for a situation in which the measurement accuracy of multiple sensors is similar. If there are many sensors with poor measurement accuracy in the system, the fusion results will be affected greatly. In general, once the number of sensors with a larger error exceeds the number of sensors with a smaller error or individual sensors are seriously interfered with, the accuracy of fusion results will decline and be even lower than that of single sensor measurement.

Therefore, a global Kalman filter was constructed, as reported in this paper, considering the contribution of each independent Kalman filter. It takes the initial observation value of each sensor and the arithmetic average of the independent filtering results as the input, so the measurement variance of each sensor can be obtained, which paves the way for further calculation of the final fusion weights.

The estimated results ${\hat{x}}_{i_{k}}$ of each sensor are averaged arithmetically. Then, the Kalman filter is applied to estimate the arithmetic mean value $\bar{{\hat{x}}_{k}}$ to obtain the state value ${\hat{x}}_{i_{k}}^{new}$ of the second correction.
(19)$$\bar{{\hat{x}}_{k}}=\frac{1}{n}\sum_{i=1}^{n}{\hat{x}}_{i_{k}}$$
(20)$${\hat{x}}_{i_{k}}^{new}=\bar{{\hat{x}}_{k}}+K_{i_{k}}^{new}\left(z_{i_{k}}-H\bar{{\hat{x}}_{k}}\right)$$

The result of the global Kalman filter can be used as a standard value to calculate the measurement variance of each sensor, which provides a basis for further weights allocation.

#### 3.3.3. Measurement Variance Estimation

The purpose of this stage was to obtain the accurate measurement variance of each sensor as a basis for calculating the final fusion weights. Considering the internal noise of the sensors and environmental interference plus other factors, this paper used the combination of instant variance and previous variance to calculate the final measurement variance. The use of instant variance enhances the sensitivity to environmental interference and previous variance emphasizes the influence of sensor factors on the measured value.

This paper introduces the forgetting factor α to enhance the amount of information provided by new data and consider the influence of old data to calculate the final measurement variance $\sigma_{i_{k}}^{2}$ as shown in Equation (22).
(21)$${\tilde{\sigma }}_{i_{k}}^{2}={\left[{\hat{x}}_{i_{k}}^{new}-H\bar{{\hat{x}}_{k}}\right]}^{2}$$
(22)$$\sigma_{i_{k}}^{2}=\alpha {\tilde{\sigma }}_{i_{k}}^{2}+\left(1-\alpha \right){\bar{\sigma }}_{i_{k}}^{2}$$
where ${\tilde{\sigma }}_{i_{k}}^{2}$ is the instant variance of measurement of sensor i at the time k. The calculation of historical measurement variance ${\bar{\sigma }}_{i_{k}}^{2}$ is the mean of measurement variance in the past period. However, this kind of artificial weight calculation method will inevitably lead to the loss of error characteristics of all sensors, which will greatly reduce the contribution of high-precision data of the sensors to the final fusion results. Considering the continuity of the measurement variance of the sensors, this paper proposes to use a NN with memory to estimate the future measurement variance by considering the previous m times. Then, it fuses the predicted measurement variances with the real-time measurement variance through the forgetting factor α.
(23)$$\alpha =\frac{m}{m+1}$$
where m is the historical step size of the prediction input.

The basic NN can only deal with a single input, but the former input and the latter input are not related at all. The one-way non-feedback connection mode of the network means it can only deal with the information contained in the current input data, while the data outside the current time period have no contribution. However, the common time series data are related closely, especially in the process of tracking. Because of the performance characteristics of the aircraft, the position state of its continuous flight time is such time-series data. Because basic recurrent neural networks (RNN) can deal with certain short-term dependence and cannot deal with long-term dependence, this research selected the LSTM model, which is more suitable to solve this problem and hardly encounters gradient dispersion. The data of the time m+1 can be predicted by inputting the historical data of the previous m times, which is a typical LSTM univariate prediction.

The key of LSTM in solving the problem of time-series data is that the state of the hidden layer will save the previous input information. Based on neuron sequential connection, it also adds a neuron feedback connection. It not only considers the input of the current time, but also gives the network a memory function for the previous information. When it propagates forward, its input layer and the information from the hidden layer of the previous time act on the current hidden layer together. In contrast to RNN, LSTM introduces three ‘gate’ structures and long- and short-memory in each neuron as shown in Equations (24)–(28). Through the control of three gates, the historical information can be recorded, so as to have a better performance in a longer sequence, referred to Appendix B.

The ‘forget’ gate needs to determine how much of the unit state $C_{t-1}$ at the previous time is retained to the current time when information enters the LSTM network which conforms to the algorithm authentication and cannot be forgotten. The forgetting gate of the inclusion layer determines whether the unit state at the previous time can be retained to the current time by substituting the input sum into Equation (24).
(24)$$f_{t}=\sigma \left(W_{f}\cdot \left[h_{t-1},x_{t}\right]+b_{f}\right)$$

The ‘input’ gate determines how many network inputs are saved to the memory unit state at the current time through sigmoid. Secondly, the layer will generate a vector to update the state of the memory unit. Then, data obtained through the forgetting gate and the input gate are substituted into Equations (25) and (26), so that the current unit state and the long-term unit state can be combined to form a new unit state.
(25)$$i_{t}=\sigma \left(W_{i}\cdot \left[h_{t-1},x_{t}\right]+b_{i}\right)$$
(26)$${\tilde{C}}_{t}=\tan h\left(W_{C}\cdot \left[h_{t-1},x_{t}\right]+b_{C}\right)$$

Equation (27) is the cell state (long time) at time t.
(27)$$C_{t}=f_{t}\cdot C_{t-1}+i_{t}\cdot {\tilde{C}}_{t}$$

The ‘output’ gate uses the current unit state $C_{t}$ to control the output of $h_{t}$ of LSTM. It obtains an initial output through the sigmoid layer as shown in Equation (28). Then, it pushes the cell state value between −1 and 1 by layer. Finally, the output of the sigmoid layer and other layers is multiplied to filter the unit state. The model output is obtained as shown in Equation (29).
(28)$$o_{t}=\sigma \left(W_{o}\cdot \left[h_{t-1},x_{t}\right]+b_{o}\right)$$
(29)$$h_{t}=o_{t}\cdot \tan h\left(C_{t}\right)$$
where $f_{t}$ is the result parameter of the forget gate. σ(⋅) indicates sigmoid activation function. Tanh(∗) indicates tanh activation function. $W_{f},W_{i},W_{C},W_{o},$ and $b_{f},b_{i},b_{C},b_{o}$ are parameters which need to be trained. ${\tilde{C}}_{t}$ is the output of the input gate. $o_{t}$ is the process parameter of the output gate. $x_{t}$ is the input time t data. In the LSTM-GWFA method of this paper, the measurement variance of continuous time m is used as the input of the LSTM network to predict the value at time m+1.
(30)$${\bar{\sigma }}_{i_{k}}^{2}=LSTM\left(\left[\sigma_{i_{t-m}}^{2},\cdots \sigma_{i_{t-1}}^{2},\sigma_{i_{t}}^{2}\right]\right)$$

#### 3.3.4. Data Weighted Fusion

Through the accurate measurement variance obtained in the previous stage, the fusion weight is calculated using Equation (12). In the process of final data fusion, this paper used the results of independent sensor filtering as the input, but did not use the original sampling data directly so that the result of data fusion is more reliable.

A weight elimination mechanism was proposed. When the value of the assigned weight is less than a certain threshold, the value of the weight will be set to zero and its value will be allocated to other weights in proportion. However, this method is not limited to the number of sensors, so as to ensure that this method cannot trigger weight truncation.
(31)$$\sigma_{avg}{}^{2}=\frac{\sum_{i=1}^{n}\sigma_{i}{}^{2}}{k}\;,\sigma_{i}{}^{2}<\sigma_{base}{}^{2}$$
(32)$$\{\begin{array}{c} \sigma_{i\_new}{}^{2}=\sigma_{i}{}^{2}+\sigma_{avg}{}^{2}\;,\sigma_{i}{}^{2}\geq \sigma_{base}{}^{2}\\ \sigma_{i\_new}{}^{2}=0\;,\sigma_{i}{}^{2}<\sigma_{base}{}^{2} \end{array}$$
(33)$$w_{i\_new}=\frac{1}{\sigma_{i\_new}{}^{2}}$$
(34)$$x_{fusion}^{k}=\frac{\sum_{i=1}^{n}w_{i\_new}{\hat{x}}_{i_{k}}}{\sum_{i=1}^{n}w_{i\_new}{\hat{x}}_{i_{k}}}$$
where $\sigma_{avg}{}^{2}$ is the truncated mean measurement variance. $\sigma_{base}{}^{2}$ is the threshold of measurement variance truncation. $\sigma_{i\_new}{}^{2}$ is the updated measurement variance. Is the updated fusion weight. $x_{fusion}^{k}$ is the final fusion result.

## 4. Simulation Results and Analysis

In this section, the accuracy of the proposed algorithm is verified and evaluated by analyzing the fusion results of sensors on the trajectory of UAVs. In the numerical simulation, the proposed method will be compared with other data fusion methods to verify the performance and the effectiveness of the algorithm will be verified in the equivalent physical simulation. The evaluation of the algorithm was mainly carried out from these several parameters: root mean square error (RMSE), mean absolute error (MAE), mean absolute percentage error (MAPE), and index of agreement (IA). These four kinds of measurement indexes with their formulas are shown in (35)–(38). n represents the number of samples, $y_{i}$ represents the observed value, and $f_{i}$ represents the predicted value [38].

The root mean square error (RMSE) is based on the mean square error, and it can be used to measure the deviation between the predicted value and the observed value.
(35)$$\delta_{RMSE}=\sqrt{\frac{1}{n}\sum_{t=1}^{n}{\left(f_{i}-y_{i}\right)}^{2}}$$

Mean absolute error (MAE) refers to the average of the absolute value of the error between the predicted value and the observed value. The average absolute error can avoid the cancellation of the positive and negative errors, so it can better reflect the actual error size.
(36)$$\delta_{MAE}=\frac{1}{n}\sum_{t=1}^{n}\left|f_{i}-y_{i}\right|$$

Mean absolute percentage error (MAPE) is usually a statistical indicator that measures the accuracy of forecasts, such as time-series forecasts. The smaller the value of MAPE, the higher the accuracy of the model.
(37)$$\delta_{MAPE}=\frac{100}{n}\sum_{t=1}^{n}\left|\frac{f_{i}-y_{i}}{y_{i}}\right|$$

Index of agreement (IA) describes the difference between the real value and the predicted value. The larger the value, the more similar the fusion value is to the real value.
(38)$$I_{IA}=\frac{\sum_{t=1}^{n}{\left|f_{i}-y_{i}\right|}^{2}}{\sum_{t=1}^{n}{\left(\left|f_{i}-\bar{y}\right|+\left|\bar{y}-y_{i}\right|\right)}^{2}}$$

### 4.1. Algorithm Numerical Simulation

The simulation program was written in the MATLAB simulation environment which ran in an Intel (R) Core (TM) i7-9700k, 32G RAM, and 64-bit Windows 10 operating system. In this paper, the LSTM model of this algorithm has two hidden layers and the number of nodes in the hidden layer is 10. The characteristic dimension of the single input neural network is 3, since the prediction of measurement variance is a variable prediction problem and the output dimension is 3. Batch processing parameter selection is 100. The same dataset was used for 1000 iterations of cyclic training. The learning rate of the neural network training is 0.001. Tanh function is selected as the activation function of the hidden layer. The output layer activation function is Softmax. The training step length of the LSTM is 10. The mean square error is regarded as the loss function. Then, the back propagation and gradient descent algorithm with adaptive learning rate are used to optimize the parameters iteratively until the loss function converges to complete the model training. The set of simulation parameters is as shown in Table 1.

All the operation data of the tracking aircraft of the four sensors were collected by the mean sampling method and the sampling frequency of each associated feature was unified as 0.5 s. If there were missing data in the sampling time period of a sensor, the data of the previous time were used to make up the missing value. Due to the large difference in the value range of each measurement point, the minimum and maximum normalization method was used to normalize the features of each dimension to eliminate the impact of the difference in value range on the accuracy of the model. We obtained 871 times track sampling data and the column data of each track were different. The data in the dataset were a three-dimensional feature, then 671 pieces were used as a training set, 100 pieces were used as a verification set, and 100 pieces were used as a test set. We considered the comparison with various existing advanced algorithms, and compared the final 100 data fusion results with CNN-LSTM [39], KF-GBDT-PSO [40], FCM-MSFA [41], and MHT-ENKF [42]. CNN-LSTM was used to predict the trajectory of sick gait children in the original reference. This paper modified the training content of CNN-LSTM as speed and acceleration, with a sliding window from 5 to 10, which matches the task of this study. KF-GBDT-PSO was used for multisource sensor data fusion to improve the accuracy of vehicle navigation in original reference. This paper took the KF-GBDT-PSO as the output of the prediction model directly and fused it with real-time observation directly as the final result. FCM-MSFA is a weighted integral data fusion algorithm for multiple radars which was proposed to reduce the error of observation trajectory and improve the accuracy of the system in the original reference. This paper ignored the correction action of FCM-MSFA on radar and retained the interference of acceleration speed, regarded as an interference input with random characteristics. MHT-ENKF is a modified ensemble Kalman filter (EnKF) in the multiple hypotheses tracking (MHT) to achieve more accurate positioning of the targe with the high nonlinearity in original reference. This paper used the data process of a single target trajectory of MHT-ENKF and ignored the trajectory correlation part, which the result as prediction of a sensor. Then, this paper adjusted the data acquisition frequency of all algorithms to 0.5 s.

In order to compare the advantages of the components of LSTM-GWFA, six comparison algorithms were added, which are N-N, N-GWFA, LSTM-N, N-AWFA, 5S-NN-GWFA, and 10S-NN-GWFA. The algorithm of N-N was used to remove the stages of LSTM and GWFA and preserve the rest in the algorithm of LSTM-GWFA. The algorithm of N-GWFA was used to remove the stage of LSTM and preserve the rest in the algorithm of LSTM-GWFA. The algorithm of LSTM-N was used to remove the stage of GWFA and preserve the rest in the algorithm of LSTM-GWFA. The above three algorithms were for ablation analysis to demonstrate their advantages. The algorithm of N-AWFA was used to replace the global Kalman filter with a mean value in the algorithm of N-GWFA. The algorithms of 5S-NN-GWFA and 10S-NN-GWFA were used to replace the stage of LSTM with neural networks (NN) with different training prediction steps (5 and 10) in the algorithm of LSTM-GWFA. The above three algorithms were for optimization and comparison of LSTM-GWFA algorithm components. Then, we made a simple analysis for added algorithms by RMSE and IA and used Equations (35)–(38) to compare the statistical results as shown in Figure 3, Figure 4, Figure 5 and Figure 6.

Figure 3 shows that the RMSE of LSTM-GWFA of this paper was the smallest. In particular, other algorithms made the results closer to high-precision radar results and the LSTM-GWFA algorithm improved the overall accuracy, which produced errors smaller than the errors of high-precision radar. CNN-LSTM is also a fusion algorithm based on LSTM and has high accuracy improvement so that low-precision radar data would not have a great impact on tracking, but the fusion result of CNN-LSTM is not as good as that of high-precision radar and its error fluctuates greatly. The error stability of the algorithm in this paper is high, which is due to the stage of weight fusion using the filtering results but not the original observation data by adaptive weights referred to in Equation (33). Figure 4 and Figure 5 show the error stability of each algorithm in each track. When the detection error of each sensor fluctuates greatly, the LSTM-GWFA algorithm in this paper can still maintain good error stability, in contrast to other algorithms that will be affected. Figure 6 shows the similarity between each fusion track and the real track. CNN-LSTM, KF-GBDT-PSO, and LSTM-GWFA algorithms all showed a high degree of approximation (IA is less than 0.1), but the LSTM-GWFA algorithms had good stability and IA was close to zero, almost matching the real track.

Figure 3, Figure 4, Figure 5 and Figure 6 also show the performance of each component of the LSTM-GWFA. The N-N algorithm without LSTM and GWFA did not achieve any improvement, and its stability and accuracy of fusion were the worst. Even the fusion results exceeded the sensor observation, with the lowest accuracy affected by the iterative cumulative error. On the basis of the N-N algorithm, the performance of the N-LSTM algorithm after adding GWFA can be significantly improved, but the performance decreased after replacing AWFA, which shows that GWFA is more adaptive to filtering than the given mean value. When the stage of GWFA was removed based on the LSTM-GWFA algorithm, the performance of LSTM-GWFA was only slightly weakened, which was equivalent to or even better than that of CNN-LSTM. On the other hand, the base line had great performance improvement after adding the stage of LSTM, which was enough to show that LSTM is the core part of the architecture. Therefore, we further replaced the LSTM network with the traditional NN with different training steps for testing. The performance of 5S-NN-GWFA and 10S-NN-GWFA were fragile compared with that of LSTM, but it could be seen that the alternative network of short-step training was significantly better than that of long-step training in the fusion architecture. From the overall algorithm comparison, FCM-MSFA and MHT-ENKF as traditional filtering algorithms had almost no difference in fusion performance. CNN-LSTM and KF-GBDT-PSO algorithms were improved by adding the NN algorithm, but their curve trend of RMSE was similar, indicating that the network structure had not changed, which affected the fusion performance. The LSTM-GWFA in this paper also used NNs combined with the measurement variance of real-time data, but not all of them. The improvement of the efficiency and stability of the algorithm also benefited from the motion maneuver detection filter and adaptive truncation weight, which not only greatly reduced the error of the fusion results but also made the accuracy of all individual radars high.

Table 2 shows the statistical results of the fusion data of 100 test tracks, in which the LSTM-GWFA algorithm shows high performance in mean square deviation. Compared with the advanced CNN-LSTM algorithm, the accuracy of the LSTM-GWFA algorithm was improved by 66% and the matching degree with the real tracks was improved by nearly 10 times. Although the algorithms of NN, N-GWFA, and N-AWFA show good fusion stability, there is still a large error above 25 with the real results. The algorithms of LSTM-N prove that the LSTM network is an indispensable part of the algorithm structure of this paper, which improved the overall performance by nearly six times. Moreover, the performance of algorithms of 5S-NN-GWFA and 10S-NN-GWFA may have been slightly poor and affected by the lack of time correlation, but it still had a certain improvement compared with the traditional filtering algorithm. Computer simulation results show that the improved adaptive weight fusion algorithm not only inherits the advantages of all traditional method algorithms but also has stronger noise suppression ability, higher data smoothness, and higher fusion precision than the advanced NN algorithm.

### 4.2. Equivalent Physical Simulation

In order to further verify the application effect of the LSTM-GWFA algorithm in air tracking, this paper used a mature refitted civil wing UAV platform which was fixed for equivalent flight test by adding detection sensors and task processors. Then, the algorithm verification was carried out by transplanting the algorithm program of adaptive data fusion which was completed in a fast, safe, low-cost, and multiple sorties manner. The flight test system consisted of refitted civil UAVs and a set of ground station equipment as shown in Figure 7.

In terms of information exchange, the air ground link between the ground station and UAVs could monitor the status of all UAVs and send simulation tasks. UAVs broadcast their position, speed, attitude, and other navigation information to the link, which gave the ground stations complete global information and the same level of autonomy and decision-making ability. The performance of UAVs and ground station equipment are shown in Table 3. The performance of the detection sensor carried by the UAVs is shown in Table 4.

Considering that there may be many initial encounter scenarios in air combat, this paper designed five single UAV flights with the same sorties of sensors and the data fusion trajectory tracking test with different initial numbers of 2, 3, 4, 5 sensors, shown in Figure 8, Figure 9 and Figure 10. The accuracy of each sensor was different, but within the performance range described in Table 4. The initialization parameters and statistical results of the tests are shown in Table 5. In this paper, the data fusion results were counted within the effective sensor detection time range to study the effectiveness and stability of the algorithm in practical applications.

In the off-site flight test, we could not obtain the real position information of the aircraft in the ideal environment directly, but we used the high-precision data of GPS as the standard value of the real position, and the detection data using longitude, latitude, and altitude as the comparison parameters of the data fusion components refer to practical application in air.

Figure 8 and Figure 9 show that data fusion of different accuracy and number of sensors can reduce the error of tracking, which was significantly better than the detection and tracking of a single sensor under the LSTM-GWFA algorithm. Especially when the noise error of individual sensors was very large, from 887 s to 894 s, high-precision fusion could also be achieved to suppress the interference of large error measurement data. Figure 10 shows that the tracking accuracy was within 10 m and the stable tracking accuracy within 1 m was achieved from 897.8 s under the five sensors in the height data fusion. This indicates that the maneuvering detection converges while the flight altitude was almost unchanged, so the fusion effect may be further improved with the increase of the number of sensors. Further, we still used the error data statistics of absolute distance as the standard for the actual application performance of the algorithm, as shown in Table 5.

Since the GPS data will also have errors, we removed the IA data statistics and added the error reduction rate (ERR) compared with low precision sensors. Table 5 shows that the data fusion effect of five sensors was the best; the fusion effect of two sensors was almost the same but in MAPE. This phenomenon occurs mainly because a high-precision filtered data is directly used as the final fusion result in the fusion process of two sensors, while ignoring the impact of low-precision sensors. It will lead to the stability of the overall effect. However, the deviation degree of the error was still the largest in data fusion of two sensors from RMSE and MAE. Table 5 also shows that the fusion effect also improved with the increase of the number of sensors and the increase of the number of low-precision sensors had no impact on the fusion results. This case illustrates the LSTM-GWFA algorithm only suppressed the low-precision measured values, but not a certain sensor, which gives the method generalization ability.

Figure 11 shows the fusion trajectory results of five sensors and that the deviation of the global Kalman filter results is very large when data fusion of tracking is started, especially in maneuvering. However, with the correction of the fusion results, the deviation can be quickly corrected. In the process, the result of global filtering also reverses the fusion accuracy, making the prediction results of LSTM more accurate. After a period of time, the multisensor fusion results show stable tracking of the trajectory, which proves the effectiveness of the method and has certain universality for the conventional motion model.

## 5. Conclusions

With the rapid development of information society, intelligence has entered every corner of our life. Intelligent control needs to process a large number of sensor data. Because a sensor itself may have unpredictable failure problems, multisensor data fusion technology came into being. Aiming at the problem of accurate trajectory tracking for dynamic targets in air combat, this paper proposes an adaptive data fusion architecture by analyzing the operational application scenarios and summarizing the existing data fusion trajectory tracking methods. Based on the traditional multisensor weighted fusion algorithm, considering the weak correlation of a single Kalman filter, the data fusion method of global Kalman filter and LSTM prediction measurement variance was designed, and the adaptive truncation mechanism was used to determine the optimal weights. At the same time, considering the accurate tracking of target with maneuvering, a maneuvering detection mechanism was added to the filter which results were used as data fusion parameters. The effectiveness and stability of the data fusion tracking method for tracking were verified by numerical simulation and field experiments, and it has certain generalization ability.

From the simulation verification, the LSTM-GWFA method has certain advantages compared the existing more advanced methods. It can not only reduce the fusion error, but also correct the comprehensive error in most cases. The fusion result is smaller than the measurement results error of the highest-precision sensor. From the off-site flight test, this paper found the data fusion results of the LSTM-GWFA method unable to achieve accurate filtering during initial tracking, but it will be stable in a short time, and with the increase of the number of sensors, the performance of the LSTM-GWFA also improved. Notably, when the number of sensors was changed from 2 to 5, the error improvement rate reached 90.93%. Even so, there are still many problems worthy of further exploration in the research process.

Due to the imperfection, redundancy, and correlation of multisensor fusion data, it needs to more accurately and comprehensively describe the actual situation of the measured object, so as to make more correct judgments and decisions. In recent years, the use of artificial NN to deal with multisensor data fusion has gradually become a research direction for scholars in various countries. However, while the reasonable application of NN in data fusion gives the fusion algorithm strong robustness and generalization, it is still a difficult problem. This is the follow-up research guided by this paper. The NN will be improved to adapt to the data fusion trajectory tracking of various moving targets.

Again, tracking a dynamic target in air combat is an important link in the process of attacking the target. Theoretical research should provide support for practical applications to deal with a variety of emergencies, such as sensor data loss, transmission signal interruption, sensor time and space alignment, heterogeneous sensor data unification, and so on. On the basis of this paper, there are a lot of problems in the field of multisensor data fusion that need to be further studied, which is bound to become the trend of future exploration.

## Figures and Tables

**Figure 1 sensors-22-05800-f001:**
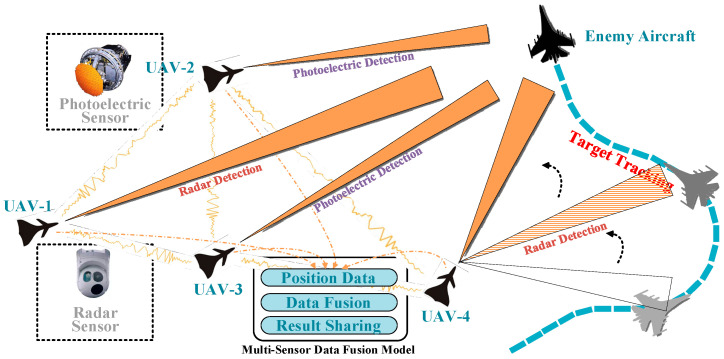
The scene of multisensor data fusion for tracking a maneuvering target in air combat.

**Figure 2 sensors-22-05800-f002:**
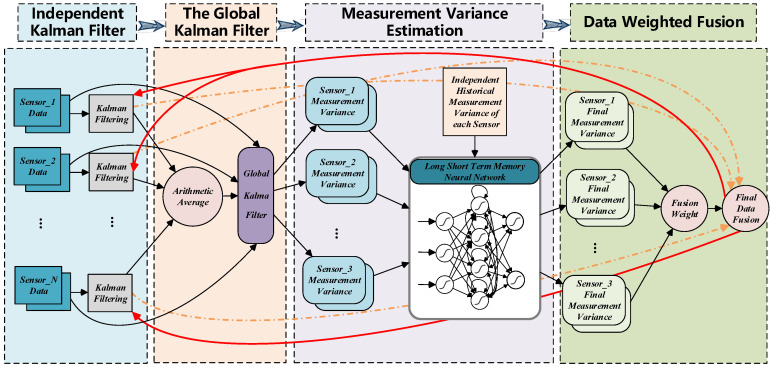
The operation process architecture of LSTM-GWFA.

**Figure 3 sensors-22-05800-f003:**
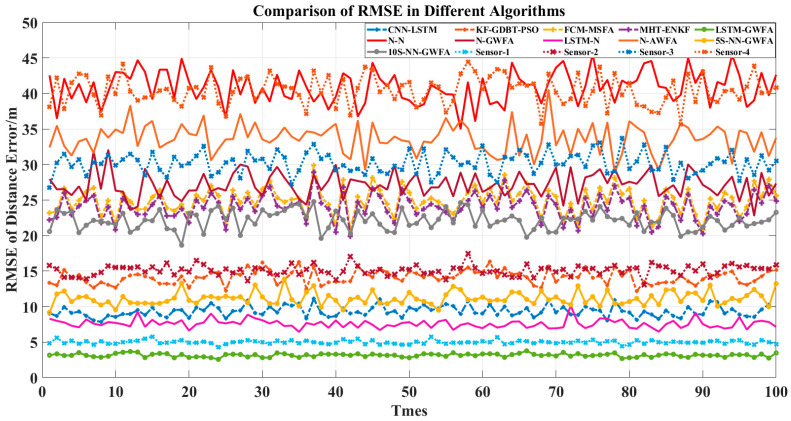
Comparison of RMSE with 100 tracks in different algorithms.

**Figure 4 sensors-22-05800-f004:**
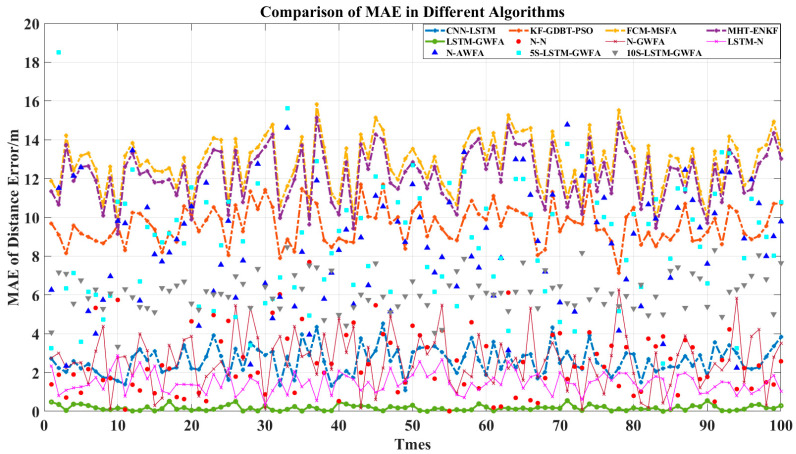
Comparison of MAE with 100 tracks in different algorithms.

**Figure 5 sensors-22-05800-f005:**
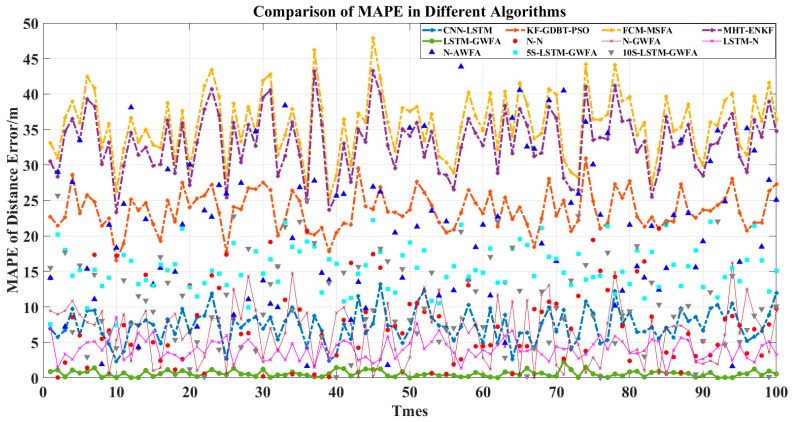
Comparison of MAPE with 100 tracks in different algorithms.

**Figure 6 sensors-22-05800-f006:**
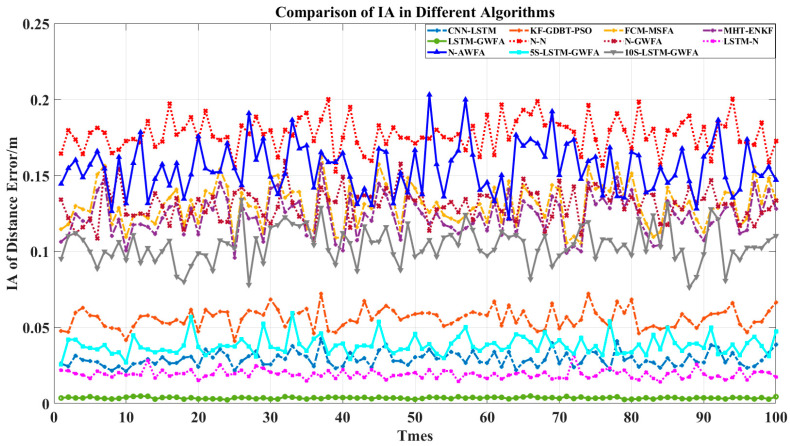
Comparison of IA with 100 tracks in different algorithms.

**Figure 7 sensors-22-05800-f007:**
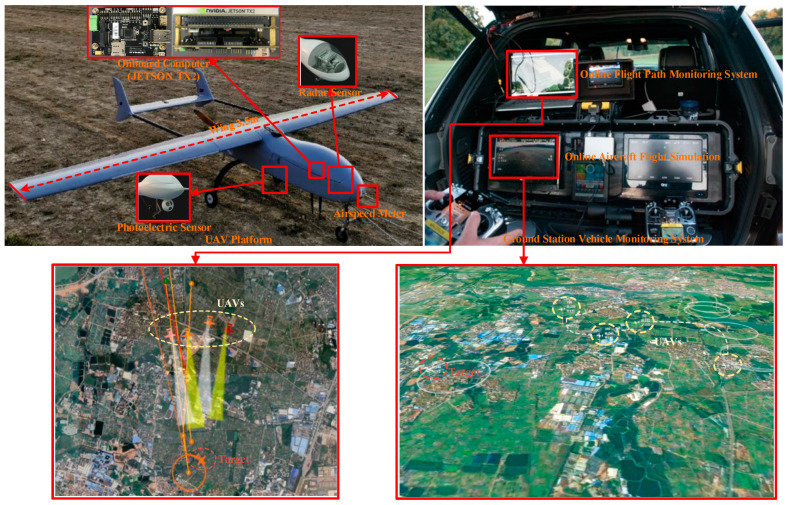
Composition of flight test system.

**Figure 8 sensors-22-05800-f008:**
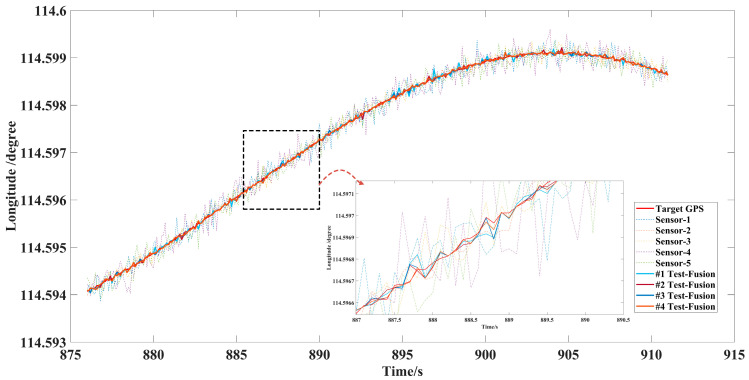
Longitude error of multisensor data fusion target trajectory.

**Figure 9 sensors-22-05800-f009:**
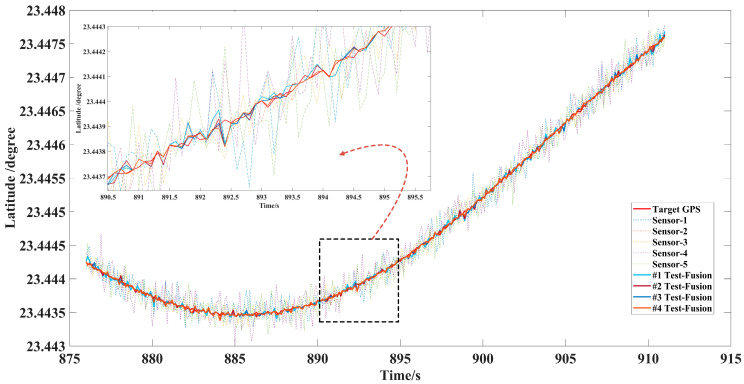
Latitude error of multisensor data fusion target trajectory.

**Figure 10 sensors-22-05800-f010:**
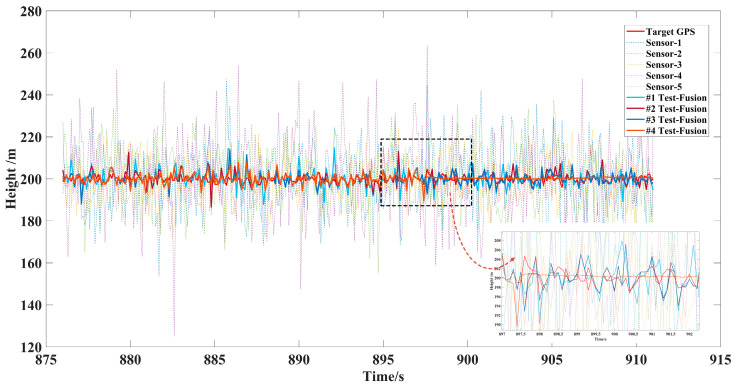
Height error of multisensor data fusion target trajectory.

**Figure 11 sensors-22-05800-f011:**
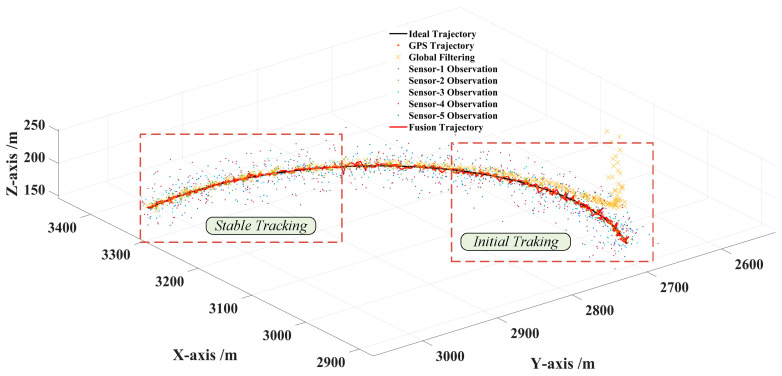
Three-dimensional schematic diagram of tracking results by five sensors.

**Table 1 sensors-22-05800-t001:** Important hyperparameter settings.

Parameter Name	Parameter Description	Value
*batch_size*	*Batch size*	100
*learning_rate*	*Initial learning rate*	0.001
*epoch*	*Number of iterations*	1000
*s1_num*	*Number of neurons in characteristic fusion layer*	16
*lstm_num*	*Number of hidden units in LSTM layer*	128
*s2_num*	*Number of hidden units in output transformation layer*	1
*m*	*Time series step size*	5

**Table 2 sensors-22-05800-t002:** Statistical results with 100 tracks in different algorithms.

Algorithm Name	Mean Square Deviation			
	RMSE	MAE	MAPE	IA
*CNN-LSTM*	9.3293	2.7235	7.7508%	0.0295
*KF-GBDT-PSO*	14.1762	9.6664	23.7586%	0.0565
*FCM-MSFA*	24.9771	12.8531	36.0786%	0.1315
*MHT-ENKF*	24.0713	12.2834	33.4910%	0.1238
*LSTM-GWFA*	**3.1672**	**0.2215**	**0.7437%**	**0.0037**
*N-N*	40.9646	2.9509	9.45811%	0.1777
*N-GWFA*	27.2367	2.9698	7.54718%	0.1311
*LSTM-N*	7.61598	1.5690	4.30846%	0.0195
*N-AWFA*	32.7377	9.0784	23.6505%	0.1562
*5S-NN-GWFA*	10.1877	9.2575	15.7572%	0.0388
*10S-NN-GWFA*	23.1720	6.2006	12.4955%	0.1047

**Table 3 sensors-22-05800-t003:** The performance of UAV and ground station equipment.

UAV Performance	Parameter Value	Function of Ground Station
*Endurance Time*	≥2 h	① *UAV Flight Area Control (Range: 5 km × 5 km)* ② *Equivalent enemy aircraft speed control (UAV Speed 30 m/s)* ③ *UAV position information receiving* ④ *UAV speed information receiving* ⑤ *UAV flight altitude control and monitoring of UAV* ⑥ *UAV initialization information input, etc.*
*Flight Altitude*	300~3500 m	
*Cruise Speed*	28~36 m/s	
*Maximum Flight Speed*	44 m/s	
*Stall Speed*	19.4 m/s	
*Turning Radius*	≤400 m	
*Maximum Climbing Speed*	≤2.5 m/s	
*Maximum Descent Speed*	3.5 m/s	
*Engine*	3W-342i (24 KW)	
*Airborne Power*	400 W (Single battery power supply 48 V, Battery capacity 20,000 mah)	

**Table 4 sensors-22-05800-t004:** The performance of sensors carried by UAVs.

Performance Name	Radar Sensor	Photoelectric Sensor	Functions
*Weight*	≤30 kg	≤17 kg	Detect, Identify, Locate and Track Air Targets (UAVs). Align time and space to obtain the position of the aircraft.
*Power*	≤200 W	≤200 W	
*Operating Frequency*	16 GHz	16 GHz	
*Tracking Distance*	≥5 km	≥5 km	
*Search Field*	±30°	/	
*Ranging Accuracy*	≤20 m	≤25 m	
*Angle measurement accuracy*	0.5°	0.2°	
*Search Angular Velocity*	/	≥60°/s	
*Minimum Target Recognition*	RCS: 5 m^2^	Light: 3 m × 3 m	

**Table 5 sensors-22-05800-t005:** The error data statistics of absolute distance.

Test Number of Fight Data Fusion	Initialization Parameters		Data Fusion of Single Trajectory			
	Number of Sensors	Ranging Accuracy (m)	RMSE	MAE	MAPE	ERR
**#1 Test**	$N_{R}=1$ $N_{P}=1$	$Acc_{R\_1}=\pm 5\;m$ $Acc_{P\_1}=\pm 15\;m$	3.7815	3.0366	50.61%	72.70%
**#2 Test**	$N_{R}=2$ $N_{P}=1$	$Acc_{R\_1}=\pm 5\;m$ $Acc_{R\_2}=\pm 15\;m$ $Acc_{P\_1}=\pm 10\;m$	2.8785	2.2760	48.65%	80.22%
**#3 Test**	$N_{R}=2$ $N_{P}=2$	$Acc_{R\_1}=\pm 5\;m$ $Acc_{R\_2}=\pm 15\;m$ $Acc_{P\_1}=\pm 10\;m$ $Acc_{P\_2}=\pm 15\;m$	2.5607	2.0050	49.98%	85.89%
**#4 Test**	$N_{R}=3$ $N_{P}=2$	$Acc_{R\_1}=\pm 5\;m$ $Acc_{R\_2}=\pm 10\;m$ $Acc_{R\_3}=\pm 15\;m$ $Acc_{P\_1}=\pm 15\;m$ $Acc_{P\_2}=\pm 20\;m$	**2.2195**	**1.7387**	**50.93%**	**90.93%**

## Data Availability

Not applicable.

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
