# Peer review of "Adaptive Data Fusion Method of Multisensors Based on LSTM-GWFA Hybrid Model for Tracking Dynamic Targets"

_sensors, 2022, doi:10.3390/s22155800_

Round 1

Reviewer 1 Report

This paper delivers a data fusion method based on multi-sensor data for tracking scenario. The experiment shows that the proposed algorithms outperforms other methods. However, I do have some concerns about this paper.

(1). Is the assumptions in Section 1 valid? For example, it is almost impossible that the communication has no delay.

(2). Following the above question, what is the purpose of each assumption? I think every assumption deserves a more detailed discussion regarding its possibility and the reason behind it.

(3). There do exist some typos. For example, line 43, it should be data instead of data.

(4). Please provide high-resolution image for each Figure. For example, Figure 3-6, and Figure 8-10 are of low quality. 

Author Response

Thank the reviewer for the valuable suggestions on this paper.
Author's Reply to the Review Report
Please see the attachment.

Reviewer 2 Report

The manuscript is well written and describes an multi-sensor data fusion method for UAV tracking. However, the current manuscript requires revisions before it can be published. 

The main comments: 

1) The experiment evaluation requires revision. As shown in the Table 2 the LSTM-GWFA method has result in a major advance comparing to the baseline systems. 

The reviewer is concerned about the baseline approaches, because their performance seems too bad for this study. Were these methods typically designed for this task in their original reference? Or did the authors make certain modifications to make them work as to compare with the proposed approach, such as changing any parameters, architecture, etc? Please be specific and highlight such details in the writeup. 

2) If the reviewer has understood it correctly, the authors have designed a system with two main components, LSTM and GWFA. In this way, the ablation analysis (or its variation) of each component should be added to demonstrate their advantage. Also, what if the authors change LSTM or other models that ouput same format? Similar things apply to GWFA. Please clarify. 

3) The literature review is not very thorough. The authors have Kalman filter and data fusion methods, however, the related literaturs on LSTM (or RNN) are missing. The reviewer has done a brief literature review for the last 3 years (a more comprehensive review is required by the authors) and the following articles should be added into the reference:

Yu, Y., Si, X., Hu, C., & Zhang, J. (2019). A review of recurrent neural networks: LSTM cells and network architectures. Neural computation31(7), 1235-1270.

Sherstinsky, A. (2020). Fundamentals of recurrent neural network (RNN) and long short-term memory (LSTM) network. Physica D: Nonlinear Phenomena404, 132306. 

Wang, Y., Zhang, X., Lu, M., Wang, H., & Choe, Y. (2020). Attention augmentation with multi-residual in bidirectional LSTM. Neurocomputing, 385, 340-347.

Other comments:

1) There is no need to repeatedly start new paragraphs throughout the manuscript (especially the ones just after the equation), such as L171, L181, L185, L188, L198, L202, L205, L212, L245, L248, L252, and etc. The reviewer only list some of them but the authors are required to make revisions throughout the manuscript. 

2) The title is worded strangly (for tracking XXX??? mulit-sensor based on LSTM hybrid model???), please consider revising the title.  

Author Response

(The authors gave the same response as above.)

Round 2

Reviewer 1 Report

I do agree that the authors have taken my suggestions into consideration and  the current version is indeed improved. 

Author Response

(The authors gave the same response as above.)

Reviewer 2 Report

The reviewer thanks the authors to make such revisions. The revised manuscript does look better.

Only comment:

The reviewer likes the added ablation analysis provided in Section 4.1. However, the Figure 3-6 are not easily readable because there are too many plots shown in the same figure.

Please consider reorgainze the figures to improve the readability.

Author Response

(The authors gave the same response as above.)
